# Postoperative, but not preoperative, central corneal thickness correlates with the postoperative visual outcomes of Descemet membrane endothelial keratoplasty

Remi Moskwa[1], Florian Bloch[1], Jean-Charles Vermion[1], Yinka Zevering[1], Dimitri Chaussard[1], Alice Nesseler[1], Christophe Goetz[2], Jean-Marc Perone[1]*

**1** Ophthalmology Department, Mercy Hospital, Metz-Thionville Regional Hospital Center, Metz, France,
**2** Research Support Unit, Mercy Hospital, Metz-Thionville Regional Hospital Center, Metz, France

* jm.perone@chr-metz-thionville.fr

**Data Availability Statement:** The datasets generated during and/or analyzed during the current study are not publicly available according

## Abstract

Descemet membrane endothelial keratoplasty (DMEK) restores visual acuity in patients with progressive corneal endothelial diseases such as Fuchs endothelial corneal dystrophy (FECD). However, patients often prefer to delay the surgery as long as possible, even though outcomes are poorer in advanced FECD. A recent study proposed that preoperative central corneal thickness (CCT) of ≥625 µm associated with worse best spectacle-corrected visual acuity (BSCVA) after DMEK for FECD. Since this threshold could signal to both surgeons and patients when to perform DMEK, we further explored the relationship between CCT and BSCVA with a retrospective cohort study. The cohort consisted of all patients with FECD who underwent DMEK in a tertiary-care hospital in 2015–2020 and were followed for 12 months. Extremely decompensated corneas were not included. Relationships between preoperative CCT and BSCVA on days 8 and 15 and months 1, 3, 6, and 12 were examined with Pearson correlation analyses. Eyes with preoperative CCT <625 or ≥625 µm were also compared in terms of postoperative BSCVA. Relationships between postoperative CCT and final BSCVA were also explored. The cohort consisted of 124 first-operated eyes. Preoperative CCT did not correlate with postoperative BSCVA at any time-point. Eye subgroups did not differ in postoperative BSCVA. However, postoperative CCT at 1–12 months correlated significantly with 12-month BSCVA (r = 0.29–0.49, p = 0.020–0.001). Thus, postoperative, but not preoperative, CCT correlated with postoperative BSCVA. This phenomenon may reflect factors that distort preoperative CCT measurements but disappear after surgery. This observation and our analysis of the literature suggest that while there is a relationship between CCT and post-DMEK visual acuity, preoperative CCT measurements may not always adequately reflect that relationship and may therefore not be a reliable predictor of DMEK visual outcomes.

to French Law No. 2018-493 of June 20, 2018 on the protection of personal data (The General Data Protection Regulation (Regulation (EU) 2016/679) (GDPR: article 9), but are available from the Clinical Research Support Platform (Plateforme d'Appui à la Recherche Clinique [PARC]) of the Regional Central Hospital (CHR) of Metz-Thionville on reasonable request (email: projet-recherche@chrmetz-thionville.fr, tel: +33 3 87 17 98 82). All non-archived data is subject to daily backups while all archived data is subject to duplicate storage at two different sites. This data processing is compliant with a baseline reference methodology (MR-004) for which the CHR Metz-Thionville signed a compliance commitment on October 8, 2018.

**Funding:** The authors received no specific funding for this work.

**Competing interests:** The authors have declared that no competing interests exist.

## Introduction

In 1998–2006, the corneal transplantation field was revolutionized by the development of two posterior lamellar keratoplasty techniques, namely, Descemet stripping automated endothelial keratoplasty (DSAEK) [1] and Descemet membrane endothelial keratoplasty (DMEK) [2]. DMEK in particular is considerably superior to penetrating keratoplasty since it associates with a low risk of graft rejection, rapid visual recovery, little change in refraction, and it restores vision almost completely [3–6].

The most common indication for DMEK is Fuchs endothelial corneal dystrophy (FECD) [7–9]. This is a slowly progressing condition in which the corneal endothelial cells are lost, thus leading to corneal edema and loss of vision [10]. Due to concerns about surgery, patients often prefer to delay DMEK as long as possible. However, because FECD eventually generates scarring and a toxic inflamed ocular environment that is hostile to the graft endothelium, the visual success of grafting drops the longer surgery is postponed [11–14]. Therefore, it would be useful to determine a clinical point that could signal to both surgeons and patients when DMEK should be conducted, thereby preserving optimal outcomes. Consequently, multiple studies have sought for pre/perioperative factors that predict the clinical outcomes of DMEK [15–20]. One is by Brockmann et al., who conducted a case-series study on 108 DMEK eyes with FECD that was published in 2019: it observed that a preoperative (host eye) central corneal thickness (CCT) of <625 μm associated with better 12-month best spectacle-corrected visual acuity (BSCVA) than thicker (≥625 μm) corneas (0.05±0.07 *vs*. 0.13±0.11 logMAR, p = 0.002). The threshold of 625 μm had been determined by receiver operating characteristic (ROC) analysis after a significant correlation between final BSCVA and preoperative CCT was found (r = 0.30, p = 0.014) [19].

Thus, it is possible that this CCT threshold could help determine when to conduct DMEK, with the aim of achieving the best possible postoperative visual acuity. However, four other studies have to our knowledge examined the relationship between preoperative CCT and final BSCVA after DMEK, and three failed to find a relationship [6, 14, 15, 21] (Table 1). To address this issue further, we performed the present study assessing the influence of preoperative CCT on the postoperative BSCVA of eyes that underwent DMEK. In addition, to explain the apparent discrepancies between the findings of our and other studies (Table 1), we also explored whether postoperative CCT has a relationship with final BSCVA after DMEK.

## Methods

### Study design

This retrospective single-center cohort study was conducted in the Metz-Thionville Regional Hospital (Grand Est, France) between October 2015 and April 2021. It was approved by the Ethics Committee of the French Society of Ophthalmology (Approval No. 00008855). All procedures were conducted in accordance with the guidelines of the Declaration of Helsinki. All patients were informed before surgery that their surgery-related data might be used for research. All consented in writing to this possibility. The consent procedure was conducted in accordance with the reference methodology MR-004 of the National Commission for Information Technology and Liberties of France (No. 588909 v1).

### Patient selection and data collection

The medical records, which are maintained prospectively, were searched retrospectively for all consecutive patients who met the following inclusion criteria: (1) adult (≥18 years); (2) eye with FECD; (3) underwent DMEK alone (if the affected eye was pseudophakic) or triple-

**Table 1. Literature assessing the relationship between preoperative CCT and postoperative visual acuity after DMEK.**

| Author, year, ref | Type of study | No. eyes | Indication | Postop time point, mo | Preop CCT, μm[a] | CCT measurement method | Type analysis | Relationship between preop CCT and postop VA | Correlation between preop and postop CCT |
|---|---|---|---|---|---|---|---|---|---|
| Our study | Retro | 124 | FECD | 1, 3, 6, 12 | 626±59 | Pachymetry | UV | r = -0.06–0.02, all NS | r = 0.16–0.21, all NS |
| Machalinska 2021 [14] | Retro | 24 | FECD, PBK | 1, 3, 6, 12 | 680±90 | AS-OCT | UV | r = 0.20–0.27, all NS | - |
| Brockmann 2019 [19] | Pros | 108 | FECD | 12 | 660±84 | AS-OCT | UV | r = 0.30, p = 0.014 | r = 0.507, p<0.001 |
| Schrittenlocher 2019 [15] | Retro | 1084 | FECD | 1, 3, 6, 12 | NI | Pentacam & pachymetry[b] | UV | r = -0.08–0.08, all NS | r = 0.02–0.11, all NS |
| Schaub 2017 [18] | Retro | 160 | FECD | 3, 6, 12, 24 | 596±53 | Pentacam | UV | All correlations NS | - |
| Peraza-Nieves 2017 [6] | Retro | 393 | FECD, PBK, other | 24 | 667±92 | Pentacam | MV | Coeff = 0.0008[c], p<0.0001 | - |

[a] Expressed as mean±standard deviation.

[b] Similar results were obtained with the two methods.

[c] For every μm higher CCT, 24-month VA increases by 0.0008 logMAR.

AS-OCT, anterior segment optical coherence tomography; CCT, central corneal thickness; Coeff, coefficient; FECD, Fuchs' endothelial corneal dystrophy; mo, months; MV, multivariate; NI, not indicated; NS, not significant; PBK, pseudophakic bullous keratopathy; preop, preoperative; pros, prospective; postop, postoperative; retro, retrospective; UV, univariate; VA, visual acuity.

DMEK (cataract surgery followed by DMEK) between October 2015 and April 2021; and (4) followed for at least 12 postoperative months. Patients were excluded from the cohort if: (1) they had another ophthalmological disease that could interfere with visual acuity (*e.g.* retinal detachment, central retinal vein occlusion, severe age-related macular degeneration, or advanced glaucoma); (2) they had a history of previous corneal graft; (3) preoperative BSCVA data were missing; or (4) the patient was lost to follow-up. Eyes were excluded from analysis if: (1) they were the second operated eye in bilateral DMEK cases; or (2) the treated eye exhibited graft failure in the 12 postoperative months. Note that because we routinely use DSAEK to treat patients with very thick and extremely decompensated corneas due to long-standing bullous keratopathy, the study cohort did not include such patients.

### Preoperative and postoperative tests and data collected

All patients underwent the following tests before surgery and 8 and 15 days and 1, 3, 6, and 12 months after surgery: BSCVA (expressed in logMAR), CCT with non-contact ultrasonic pachymetry (Tono pachymeter NT-530P; Nidek Co., Gamagori Aichi, Japan), and anterior-segment optical coherence tomography (OCT) (NIDEK with a special module; Nidek Co., Gamagori Aichi, Japan). Mean anterior keratometry was also conducted before surgery (Visionix Luneau L67 auto-kerato-refractometer, Pont-de-l'Arche, France). Graft endothelial cell density (ECD) was provided by the eye bank.

The following data were collected: patient age and gender; preoperative and postoperative BSCVA and CCT; preoperative mean anterior keratometry, K1, and K2; donor age; preoperative (graft) ECD; graft origin; operative time; use of triple-DMEK; and number of rebubblings, graft rejections, Irvine-Gass syndrome cases, and graft failures requiring regrafting.

BSCVA recovery between day 8 and 3 months was calculated as (day 8 BSCVA—3 month BSCVA). Postoperative change in CCT relative to preoperative CCT was denoted Delta CCT. For example, Delta CCT Day 8 = (day 8 CCT—preoperative CCT).

## Surgical technique and postoperative care

To prevent pupillary block during and after surgery, all patients underwent iridotomy at the 6 o'clock position with Nd: YAG Laser (Laser ex-super Q; Ellex Europe, Medical Quantel, Cournon d'Auvergne, France) during a preoperative consultation. All surgeries were performed by the same experienced microsurgeon (JMP). DMEK was performed as described by Melles [2]. In all cases except two, surgery was performed under general anesthesia due to surgeon preference; in the exceptions, general anesthesia was contraindicated and surgery was conducted under locoregional anesthesia (peribulbar block using 23-gauge Atkinson cannula with a 50/50 mixture of 2% lidocaine and 7.5 mg/mL ropivacaine).

Unprepared grafts were obtained from the tissue banks in Besançon or Nancy, France. All grafts were preserved by organ culture in Eurobio organ culture medium at 31˚C. They were transported in medium with dextran and were exposed to dextran for 48 to 72 hours. All grafts had a requested ECD greater than 2200 cells/mm$^2$. The following steps have been described in detail previously [22]. Thus, the graft was prepared by the microsurgeon in the operating room just before surgery, after which the patient's cornea was prepared for transplantation and the graft was injected into the anterior chamber. After positioning the graft, a sterile air (75% of cases) or 20% sulfur hexafluoride (25%) bubble was injected into the anterior chamber. In approximately half of the surgeries, the main incision was sutured with one point of Nylon 10.0 that was secondarily buried and removed 1 month later. In the case of triple-DMEK, phacoemulsification was performed before DMEK as described previously [23]. Intracameral steroids were not applied.

In the postoperative period, all patients were placed in the supine position for the first 12 hours. All patients underwent treatment with a topical antibio-corticosteroid (Maxidrol, Dexamethasone + Neomycin Polymyxine B; ALCON, Rueil Malmaison, France) four times a day. An ophthalmic ointment (Vitamin A dulcis; ALLERGAN, Courbevoie, France) was also applied two times a day to promote healing. Two months after surgery, Maxidrol was replaced with long-term low-dose corticosteroid eye drops (FLUCON; NOVARTIS Pharma, Rueil Malmaison, France).

Patients who developed Irvine-Gass syndrome, as confirmed by anterior-segment optical coherence tomography (AS-OCT) (NIDEK; Nidek Co., Gamagori Aichi, Japan), and loss of visual acuity were treated with oral acetazolamide (Diamox®; Sanofi, Gentilly, France; one 250 mg tablet 3 times/day for one month) and NSAID (indomethacin 0.1%; Chauvin, Montpellier, France; 4 times/day for one month). Cases of allograft rejection were never observed in our series.

If AS-OCT revealed detachment of 20% or more of the graft or the detachment was threatening the visual axis [24], rebubbling was performed by injecting a sterile air bubble in the anterior chamber under topical anesthesia.

A graft was considered to have failed if edema persisted after 3 months, or nonadhesion of the graft was observed despite up to four rebubblings. In these cases, regraft with DMEK or DSAEK was conducted. Graft failure due to technical problems such as graft inversion was included in this definition.

## Statistical analyses

Since graft failure rates are an important measure of the success of DMEK, patients with graft failure were included in the cohort but then excluded from all statistical analyses. The second-operated eyes in bilateral DMEK cases were also excluded from all analyses. Continuous data were expressed as mean±standard deviation and range while categorical variables were expressed as $n$ (%). Pearson's correlation tests followed by Bonferroni correction were

conducted to determine the relationships between (i) preoperative CCT and pre/postoperative BSCVA, day 8–month 3 BSCVA recovery, or postoperative CCT and (ii) 12-month BSCVA and postoperative CCT or Delta CCT. Eyes were also divided into two subgroups according to the ROC-determined 625-μm preoperative CCT threshold reported by Brockmann et al. [19]. The subgroups were compared in terms of pre/perioperative variables, postoperative BSCVA, or day 8–3 month BSCVA recovery by Student's *t*-test or Fisher's exact test, as appropriate, followed by Bonferroni correction. All statistical analyses were performed with SAS software (version 9.4, SAS Inst., Cary, NC, USA). P-values <0.05 were considered to indicate statistical significance.

## Results

### Patient disposition in the study

During the study period, 185 eyes underwent DMEK. Of these, 24 were excluded because they had another ophthalmological disease that could interfere with visual acuity (*n* = 8; two with epiretinal membrane, two with post-uveitic endothelitis, one with severe AMD, and three with retinal vein occlusion), they had undergone corneal grafting previously (*n* = 11), preoperative BSCVA data were missing (*n* = 4), or they were lost to follow-up (*n* = 1). Moreover, 18 eyes that required a second graft during the first postoperative year due to primary graft failure and 21 second-operated eyes were excluded from statistical analysis. Thus, 124 eyes (124 patients) were included in the statistical analysis (Fig 1).

### Baseline clinical and operative characteristics of the cohort

The patients were on average 72 years old and 63% were women. The average preoperative CCT, BSCVA, and mean anterior keratometry were 626 μm, 0.66 logMAR, and 44 D,

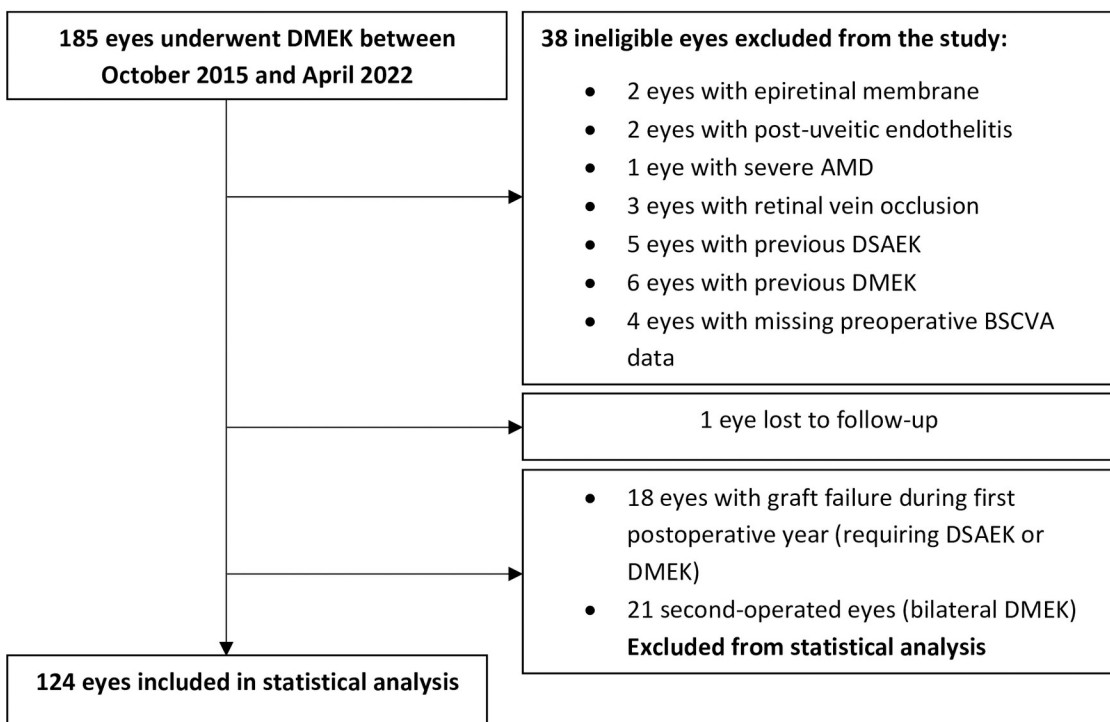

**Fig 1. Flow chart depicting the disposition of the subjects during the study.** AMD, age-related macular degeneration; DMEK, Descemet membrane endothelial keratoplasty; DSAEK, Descemet stripping automated endothelial keratoplasty.

**Table 2. Baseline and operative characteristics of the cohort (*n* = 124 eyes and 124 patients).**

| Variable | Mean±SD (range) or *n* (%) |
|---|---|
| Patient age, y | 72±9 (48–90) |
| Female sex | 78 (63%) |
| Preoperative CCT, μm | 626±59 (530–798) |
| Preoperative BSCVA, logMAR | 0.66±0.31 (2–0.2) |
| Mean anterior keratometry, D | 43.6±1.7 (36.5–48.6) |
| K1, D | 43.1±1.7 (35.8–47.9) |
| K2, D | 44.2±1.7 (37.3–49.3) |
| Donor age, y | 74±11 (30–99) |
| Graft ECD, cells/mm$^2$ | 2542±202 (2000–3040) |
| Graft origin, Besançon *vs.* Nancy | 71 (57%) *vs.* 53 (43%) |
| Operative time, min | 35±9 (20–60) |
| Triple-DMEK | 56 (45%) |
| Any rebubbling | 37 (30%) |
| Multiple rebubbling | 11 (9%) |
| Irvine-Gass syndrome | 6 (5%) |

BSCVA, Best spectacle-corrected visual acuity; CCT, Central corneal thickness; ECD, Endothelial cell density; Triple-DMEK, Phacoemulsification followed by DMEK.

respectively. The donors were on average 74 years old and the grafts had a mean preoperative ECD of 2542 cells/mm$^2$. The grafts were slightly more likely to come from Besançon rather than Nancy (57% *vs.* 43%). Triple-DMEK was conducted in 45% of the cases. DMEK surgery lasted on average 35 min (Table 2).

## Postoperative outcomes

The DMEK graft failure rate was 10% (18 of the 185 total DMEK surgeries during the study period). This reflects the learning curve of the surgeon and the fact that we switched from air to SF6 for bubbling only recently. In the last study year, our primary failure rate was 3%. Rebubbling was required in 37 of the 124 analyzed study eyes (30%), of which 11 (9% of the whole cohort) required multiple rebubblings. Six eyes (5%) presented with Irvine-Gass syndrome during follow up. All cases resolved with treatment. There were no cases of graft rejection (Table 2).

Starting 15 days after surgery, BSCVA improved markedly from 0.66 logMAR at baseline to 0.08 logMAR at 12 months. This associated with a gradual decrease in CCT from 626 μm at baseline to 538 μm at 12 months (Fig 2 and S1 Table).

## Correlations between preoperative CCT and postoperative BSCVA and CCT

Preoperative CCT did not correlate with preoperative BSCVA or BSCVA at 12 months or earlier time points. It also did not correlate with early recovery in BSCVA (defined as change in BSCVA at 3 months relative to day 8) (Table 3) or postoperative CCT (S2 Table).

## Comparison of eyes with low and high preoperative CCT in terms of postoperative BSCVA

The ROC analysis of Brockmann et al. [19] showed that a preoperative CCT threshold of 625 μm had the greatest discriminatory power. When we applied this threshold to our 124

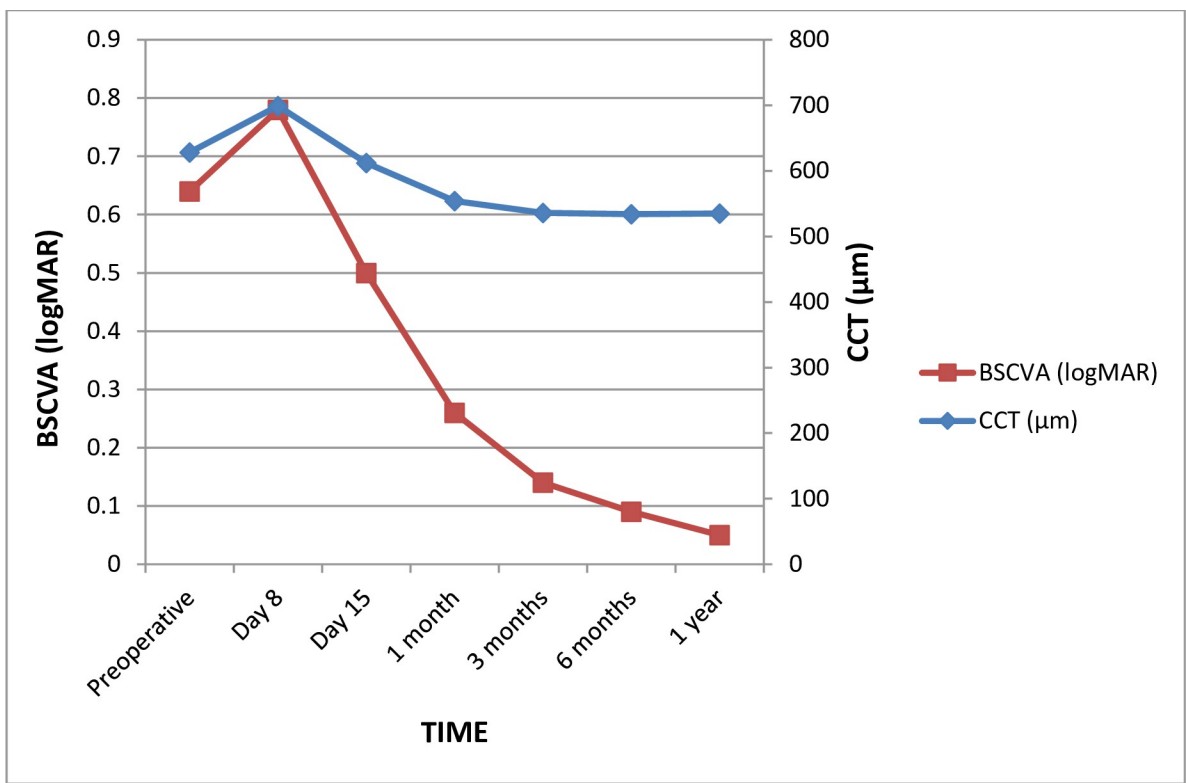

**Fig 2. Change in BSCVA and CCT After DMEK.** BSCVA, best spectacle-corrected visual acuity; CCT, central corneal thickness.

eyes, we found that 70 and 54 had preoperative CCTs of <625 and ≥625 μm, respectively. The two subgroups did not differ significantly in terms of patient or donor age, patient sex, or graft ECD. Importantly, they did not differ in either preoperative or postoperative BSCVA (Table 4 and Fig 3).

As could be expected given the lack of correlations between preoperative and postoperative CCT (S2 Table), the low and high preoperative CCT subgroups did not differ in postoperative CCT (S3 Table).

**Table 3. Correlations of preoperative CCT with postoperative BSCVA or BSCVA recovery.**

|  | Correlation coefficient r (95% CI)* | p value |
|---|---|---|
| Preoperative BSCVA | 0.20 (0.02; 0.36) | 0.207 |
| Day 8 BSCVA | -0.13 (-0.30; 0.05) | 0.999 |
| Day 15 BSCVA | -0.09 (-0.27; 0.08) | 0.999 |
| 1 month BSCVA | -0.06 (-0.23; 0.11) | 0.999 |
| 3 months BSCVA | 0.01 (-0.17; 0.18) | 0.999 |
| 6 months BSCVA | 0.02 (-0.16; 0.20) | 0.999 |
| 12 months BSCVA | 0.01 (-0.15; 0.18) | 0.999 |
| BSCVA recovery between day 8 and 3 months | 0.14 (-0.04; 0.31) | 0.911 |

* Pearson's correlation test followed by Bonferroni correction.

BSCVA, Best spectacle-corrected visual acuity; BSCVA recovery between day 8 and 3 months, change in BSCVA at 3 months relative to BSCVA at day 8.

**Table 4. Comparison of eyes with low preoperative CCT (<625 μm) and high preoperative CCT (≥625 μm) in terms of pre/perioperative variables, postoperative BSCVA, and BSCVA recovery.**

| | Low preoperative CCT (<625 μm) *n* = 70 | High preoperative CCT (≥625 μm) *n* = 54 | p value* |
|---|---|---|---|
| Patient age, y | 72±9 (48–90) | 72±8 (56–90) | 0.999 |
| Female sex | 39 (56%) | 39 (72%) | 0.770 |
| Donor age, y | 73±11 (30–91) | 75±12 (43–99) | 0.999 |
| Graft ECD, cells/mm$^2$ | 2526±182 (2180–2983) | 2562±225 (2000–3040) | 0.999 |
| Preoperative BSCVA, logMAR | 0.61±0.28 (2–0.2) | 0.72±0.33 (1.7–0.2) | 0.658 |
| **Postoperative BSCVA, logMAR** | | | |
| Day 8 | 0.83±0.45 (1.7–0) | 0.73±0.39 (1.7–0) | 0.999 |
| Day 15 | 0.51±0.43 (2--0.1) | 0.48±0.35 (1.7–0) | 0.999 |
| 1 month | 0.28±0.23 (1.3--0.1) | 0.27±0.23 (1–0) | 0.999 |
| 3 months | 0.16±0.16 (0.7--0.1) | 0.16±0.17 (0.8--0.1) | 0.999 |
| 6 months | 0.11±0.16 (1--0.1) | 0.11±0.14 (0.8--0.1) | 0.999 |
| 12 months | 0.07±0.13 (0.5--0.1) | 0.08±0.16 (1--0.2) | 0.999 |
| BSCVA recovery between day 8 and 3 months | -0.67±0.42 (0.1--1.7) | -0.57±0.36 (0--1.65) | 0.999 |

The data are expressed as mean±standard deviation or *n* (%).

*The subgroups were compared by Student's t–test or Fisher's exact test. as appropriate. followed by Bonferroni correction.

BSCVA, Best spectacle-corrected visual acuity; CCT, Central corneal thickness; BSCVA recovery between day 8 and 3 months, change in BSCVA at 3 months relative to BSCVA at day 8; ECD, Endothelial cell density.

## Correlations between postoperative CCT and 12-month BSCVA

We then explored the relationships between CCT at various postoperative timepoints and 12-month BSCVA. Notably, postoperative CCT at later time points (1, 3, 6, and 12 months) correlated significantly with 12-month BSCVA (r = 0.29, 0.45, 0.49, and 0.45; p = 0.020, 0.001, 0.001, and 0.001, respectively). However, this was not observed for CCT at days 8 and 15 (Table 5).

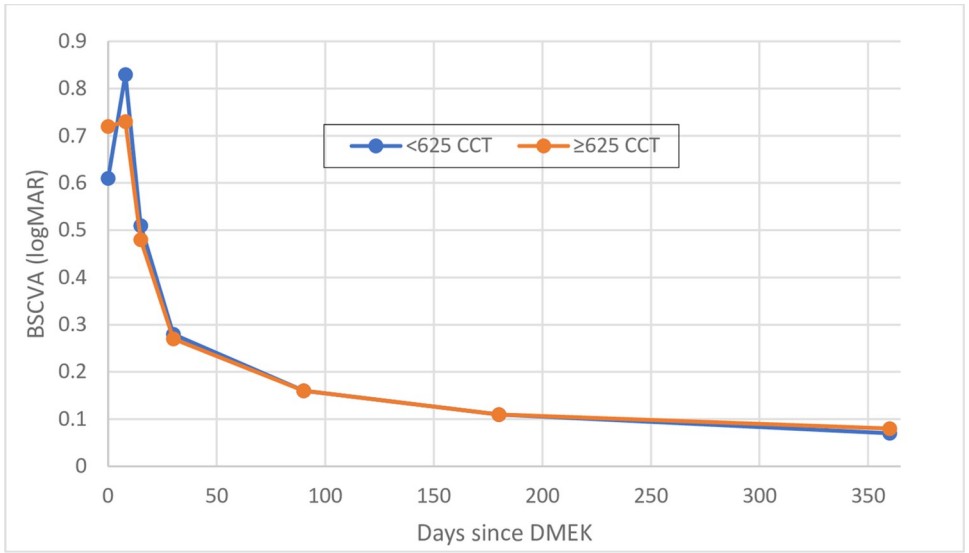

**Fig 3. Change in BSCVA in eyes with low (<625 μm) and high (≥625 μm) preoperative CCT.** BSCVA, best spectacle-corrected visual acuity; CCT, central corneal thickness.

**Table 5. Correlations of 12-month BSCVA with CCT at various postoperative timepoints.**

|  | Correlation coefficient r (95% CI)* | p value |
|---|---|---|
| Day 8 CCT | 0.04 (-0.16; 0.24) | 0.999 |
| Day 15 CCT | 0.06 (-0.14; 0.25) | 0.999 |
| 1 month CCT | 0.29 (0.12; 0.45) | **0.02** |
| 3 month CCT | 0.45 (0.3; 0.59) | **0.001** |
| 6 month CCT | 0.49 (0.34; 0.61) | **0.001** |
| 12 month CCT | 0.45 (0.3; 0.58) | **0.001** |
| Delta CCT Day 8 | 0.09 (-0.11; 0.29) | 0.999 |
| Delta CCT Day 15 | 0.09 (-0.11; 0.28) | 0.999 |
| Delta CCT 1 month | 0.21 (0.03; 0.37) | 0.264 |
| Delta CCT 3 months | 0.33 (0.16; 0.48) | **0.002** |
| Delta CCT 6 months | 0.34 (0.17; 0.49) | **0.001** |
| Delta CCT 12 months | 0.28 (0.11; 0.44) | **0.018** |

*Pearson's correlation test followed by Bonferroni correction.

BSCVA, Best spectacle-corrected visual acuity; CCT, Central corneal thickness; Delta CCT, difference between preoperative and postoperative CCT (for example, Delta CCT Day 8 = day 8 CCT minus preoperative CCT).

We also examined how change in CCT (relative to baseline; designated Delta CCT) at various time points correlated with 12-month BSCVA. Change in CCT at 3, 6, and 12 months also correlated significantly with 12-month BSCVA (r = 0.33, 0.34, and 0.28; p = 0.002, 0.001, and 0.018, respectively) (Table 5).

## Discussion

DMEK is a recently described corneal grafting technique that is used to treat diseases of the corneal endothelium. Due to the thinness of the DMEK graft, this procedure results in better visual outcomes and faster postoperative recovery relative to DSAEK and penetrating keratoplasty, which involve thicker grafts [25–27]. The present study confirmed the excellent visual outcomes of DMEK: the mean 12-month BSCVA was 0.08 logMAR, which is comparable to the visual outcomes that were recently reported by a prospective registry study on 752 DMEK eyes (0.12 logMAR at 12 months) [28].

Since corneal endothelial dystrophies are progressive diseases [29], it would be useful to identify easily measured clinical thresholds that signal when it is time to conduct DMEK. Brockmann et al. recently proposed with a case-series study on 108 eyes that a preoperative CCT of 625 μm could be such a threshold: they observed that 12-month BSCVA correlated with preoperative CCT (r = 0.30, p = 0.014) and then showed that eyes with preoperative CCTs above the ROC-determined 625-μm threshold had significantly worse 12-month BSCVA than eyes with thinner corneas [19]. This association is logical since CCT reflects the degree of corneal edema, which impairs BSCVA. However, we failed to find a correlation between preoperative CCT and 12-month BSCVA (r = 0.01, p = 0.999). Moreover, eye subgroups generated by using the preoperative 625-μm CCT threshold of Brockmann et al. [19] did not differ in final BSCVA.

We also examined whether preoperative CCT correlated with early recovery of BSCVA (defined as 3-month BSCVA minus day-8 BSCVA) because we speculated that eyes with high preoperative CCT might recover vision more slowly because the endothelial graft must resorb more intrastromal fluid. However, a correlation was not observed (r = 0.14, p = 0.911). Moreover, preoperative CCT did not correlate with BSCVA at any of the time points before 12

months. Thus, in our cohort, high preoperative CCT did not appear to reflect visual recovery and could not serve as a marker that indicates when DMEK should be conducted for FECD.

It should be noted that while postoperative CCT plateaued at 3 months, postoperative BSCVA continued to improve, albeit to a small degree. We speculate that this could reflect ongoing remodeling of the collagen matrix in the corneal stroma after the excess fluid has been pumped out by the endothelial graft; this remodeling could reduce light scattering, therefore improving BSCVA.

## Correlation between postoperative CCT and 12-month BSCVA

Our failure to find a correlation between preoperative CCT and post-DMEK BSCVA has also been observed by three other studies, including two with large cohorts (n = 160 and 1084) [14, 15, 21]. However, a study by Peraza-Nieves et al. on 393 DMEK eyes found with multivariate analysis that 1-μm higher preoperative CCT predicted 0.0008 higher logMAR BSCVA at 24 months [6]. Since their mean±standard deviation preoperative CCT was 667±92 μm, that meant an eye with 92-μm higher CCT (i.e. the average difference between the CCT of a given eye and the average CCT of all eyes) was predicted to have 0.074 logMAR higher BSCVA than average at 24 months (Table 1).

The reasons for the discrepancy between the studies of Brockmann et al. and Peraza-Nieves et al. *versus* our and the other three studies are not clear. One possibility is that the eyes in the former studies had longer standing disease and had developed stromal haze that decreased postoperative BSCVA recovery. However, the FECD stage ranges of these studies seem similar to those in our study.

Another possibility is suggested by our earlier study on the relationship between central graft thickness (CGT) and final BSCVA after DSAEK [30]. Whether thinning DSAEK grafts actually improves visual outcomes has been a vexed question in the literature, with some studies [31–36] but not others [37–44] finding an effect. Our study showed that postoperative, but not preoperative, CGT predicted final BSCVA. Therefore, we suggested that the variable CGT-BSCVA relationships reported in the literature may reflect inaccuracies or variation in the preoperative CGT measurements due to various factors (e.g. graft culture duration) that blur the relationship between CGT and visual acuity; these factors disappear after surgery, allowing the intrinsic relationship between CGT and BSCVA to emerge [30]. Our explorations in the present study suggested that a similar phenomenon may be operating in the relationship between CCT and BSCVA in DMEK: we found that unlike preoperative CCT, postoperative CCT did correlate significantly with final BSCVA. Specifically, thinner corneas in the postoperative period associated with better postoperative BSCVA. The relationship started appearing at 1 month (r = 0.29, p = 0.020) and then grew stronger and persisted through to the end of follow-up (r = 0.45–0.49, all p = 0.001). To our knowledge, this is the first time this relationship has been examined or reported.

Thus, we speculate that preoperative CCT does not always show a relationship with final BSCVA in the literature (Table 1) due to interstudy variation in the preoperative CCT measurements, which may be induced by disease- and/or preoperative setting-related factors. These factors disappear after DMEK, allowing the inherent relationship between CCT and BSCVA to become more apparent. This speculation is supported by the smaller variation in our CCT measurements at 12 months compared to our preoperative CCT measurements (standard deviation: 41 *vs.* 59 μm). It is also supported by the variable relationships between preoperative and postoperative CCT in the literature: Brockmann et al. [45] found that these two variables correlated strongly whereas we and another study that also did not find a CCT-BSCVA relationship did not observe any such correlation [46] (Table 1).

These findings together suggest that while CCT does have a relationship with post-DMEK visual acuity, preoperative measurement variation means this variable may not be a reliable predictor of visual outcomes after DMEK.

The relationship between CCT and BSCVA likely reflects the fundamental healthiness of the graft corneal endothelial cells, specifically their ability to resorb intrastromal fluid, thereby reducing the thickness of the cornea. This notion is supported by multiple *in vitro* and *in vivo* studies showing that corneal ECD dictates the efficiency of the endothelial pump [47–50]. Moreover, the Cornea Donor Study showed that postoperative CCT predicts graft failure (p = 0.002) and associates modestly with endothelial cell loss during follow-up (r = -0.29) [51]. It should be noted, however, that we did not observe a direct relationship between postoperative CCT and postoperative central ECD. Thus, when the eyes were divided according to median 6-month CCT (526 μm), the resulting two subgroups did not differ significantly in terms of 6-month ECD (1328 *vs.* 1300 cells/mm$^2$, p = 0.230 on Wilcoxon test). The same was true for 12-month CCT (median, 529 μm; 1123 *vs.* 1099 cells/mm$^2$, p = 0.440). Similarly, there were no correlations between postoperative CCT and postoperative ECD at either 6 months (r = 0.002, p = 0.981 on Spearman correlation test) or 12 months (r = 0.01, p = 0.881). Thus, while it is possible that postoperative CCT predicts postoperative BSCVA because endothelial cell function shapes CCT and CCT dictates BSCVA, it seems that the ECD variable does not adequately reflect endothelial function in our setting. It is possible that other measures of endothelial function would be able to detect this relationship.

## Study limitations

This study had some limitations. First, it was a retrospective study, which lends itself to selection and information bias. However, the data were all recorded prospectively. Second, it was a monocentric study. However, all surgeries were conducted by one surgeon, which may have limited confounding due to surgeon-related variation. Third, the sample size was relatively small (*n* = 124). However, many other studies in the field have employed such or smaller sample sizes, including that by Brockmann et al. (*n* = 108) [19]. Fourth, our study focused on moderate to severe FECD cases only: in our institution, very advanced cases that have intrastromal scars (which are irreversible structural changes and lead to poorer postoperative visual outcomes [52, 53]) are generally treated with DSAEK. Thus, our finding that postoperative BSCVA does not depend on preoperative CCT cannot be extrapolated to cases of very severe endothelial disease. Fifth, we used noncontact ultrasound pachymetry to measure CCT rather than contact ultrasound pachymetry. However, noncontact pachymetry is easily conducted during consultation and several studies have shown that these two methods differ negligibly in terms of CCT measurements [54, 55]. Sixth, since Brockmann et al. and Peraza-Nieves et al. respectively used AS-OCT and Pentacam to measure CCT [6, 45], it is possible that the CCT-measuring technique explains the different findings of our study. However, Schrittenlocher et al., who used both Pentacam and ultrasound pachymetry to measure CCT, also did not find a relationship between preoperative CCT and final visual acuity after DMEK regardless of which method they used [15] (Table 1). Moreover, while ultrasound pachymetry, AS-OCT, and Pentacam measurements of CCT are not interchangeable, they correlate closely [56–59].

## Conclusions

This study showed that postoperative, but not preoperative, CCT correlated with postoperative visual acuity outcomes, including at 12 months. Thus, patients with high preoperative CCT are not limited in terms of achieving excellent outcomes after DMEK. The fact that postoperative, but not preoperative, CCT correlated with final visual outcome may reflect greater variation in

preoperative CCT measurements relative to postoperative CCT measurements. Thus, preoperative CCT may not be a reliable indicator of post-DMEK visual outcomes.

## Supporting information

**S1 Table. Evolution of CCT and BSCVA after surgery.**
(DOCX)

**S2 Table. Correlations between preoperative and postoperative CCT.**
(DOCX)

**S3 Table. Comparison of eyes with low preoperative CCT (<625 μm) and high preoperative CCT (≥625 μm) in terms of postoperative CCT.**
(DOCX)

## Author Contributions

**Conceptualization:** Christophe Goetz, Jean-Marc Perone.

**Data curation:** Remi Moskwa, Florian Bloch, Jean-Charles Vermion, Dimitri Chaussard, Alice Nesseler.

**Formal analysis:** Remi Moskwa, Christophe Goetz.

**Funding acquisition:** Christophe Goetz, Jean-Marc Perone.

**Investigation:** Remi Moskwa, Florian Bloch, Jean-Charles Vermion, Dimitri Chaussard, Alice Nesseler.

**Methodology:** Christophe Goetz, Jean-Marc Perone.

**Project administration:** Jean-Marc Perone.

**Resources:** Christophe Goetz, Jean-Marc Perone.

**Software:** Christophe Goetz.

**Supervision:** Jean-Marc Perone.

**Validation:** Jean-Marc Perone.

**Visualization:** Yinka Zevering, Jean-Marc Perone.

**Writing – original draft:** Remi Moskwa.

**Writing – review & editing:** Remi Moskwa, Yinka Zevering, Jean-Marc Perone.

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
