## [Decision Letter · Decision Letter 0]

19 Oct 2022

PONE-D-22-27294Postoperative, but not preoperative, central corneal thickness correlates with the postoperative visual outcomes of Descemet membrane endothelial keratoplastyPLOS ONE

Dear Dr. Perone,

Thank you for submitting your manuscript to PLOS ONE. After careful consideration, we feel that it has merit but does not fully meet PLOS ONE’s publication criteria as it currently stands. Therefore, we invite you to submit a revised version of the manuscript that addresses the points raised during the review process.

We look forward to receiving your revised manuscript.

Kind regards,

Michael Mimouni

Academic Editor

PLOS ONE

Journal Requirements:

a) Did participants provide their written or verbal informed consent to participate in this study?

Reviewers' comments:

Reviewer's Responses to Questions

**Comments to the Author**

1. Is the manuscript technically sound, and do the data support the conclusions?

Reviewer #1: No

Reviewer #2: Yes

Reviewer #3: Partly

Reviewer #4: Partly

2. Has the statistical analysis been performed appropriately and rigorously? 

Reviewer #1: No

Reviewer #2: Yes

Reviewer #3: Yes

Reviewer #4: No

3. Have the authors made all data underlying the findings in their manuscript fully available?

Reviewer #1: No

Reviewer #2: Yes

Reviewer #3: Yes

Reviewer #4: Yes

4. Is the manuscript presented in an intelligible fashion and written in standard English?

Reviewer #1: Yes

Reviewer #2: Yes

Reviewer #3: Yes

Reviewer #4: No

5. Review Comments to the Author

Reviewer #1: Authors should have a plenty of data.

They should analyze and rewrite the paper.

1) There is no description of included criteria despite the prospective fashion.(Line102-107)

2)It doesn't make sense if you evaluate the preoperative factor, and postoperative factor.(Line164-174)

Please ask the specialist of statistics, and reconsider the methodology.

Please add the statistic specialist in the acknowledgement.

3)It doesn't make sense if you compare the CCT over 625, and under 625 um, since the mean CCT you showed was 695.

Please justify the use of 625. (Line 219-222)

4) The discussion explaining the correlation between postoperative CCT and the BSVCA at 12months.

You can evaluate HOAs more since you used Pentacam. Please evaluate more using HOAs.

Please refer to the following paper. (https://pubmed.ncbi.nlm.nih.gov/33052928/)

Anyway, the current version is not proper in terms of statics, or logics.

Reviewer #2: Nicely written no major concerns should add to the literature that pre op cct should not be a variable to consider. Use bcva and patient function as criteria always issues with retrospective study but not much else published

Reviewer #3: Dear Authors,

thank you very much for this interesting paper about pre- and postoperative CCT in DMEK and the influence on BCVA.

In the abstract it is stated that extremely decompensated eyes were not included. This is not present in the methods section and should be added. However, I would rather recommend to include those eyes, since this might have an impact on the prediction with preoperative CCT.

Kind regards!

Reviewer #4: In the introduction, the authors describe the recent studies that correlate prep CCT and postoperative vision. However, the primary conclusions are written in a confusing manner. Predictive factors are of course relevant when they are identified preoperatively or perioperatively. The authors should focus their analysis on the preoperative factors.

It would also be relevant to determine if postoperative CCT correlates to preop parameters.

In light of findings that preop CCT does not affect postoperative vision. Compare with other studies correlating CCT and postoperative vision such as https://pubmed.ncbi.nlm.nih.gov/30339062/, https://pubmed.ncbi.nlm.nih.gov/33591041/, https://pubmed.ncbi.nlm.nih.gov/34336256/
https://www.ncbi.nlm.nih.gov/pmc/articles/PMC8786756/or
https://pubmed.ncbi.nlm.nih.gov/27986425/

Revise accordingly.

6. PLOS authors have the option to publish the peer review history of their article (what does this mean?). If published, this will include your full peer review and any attached files.

Reviewer #1: No

Reviewer #2: No

Reviewer #3: No

Reviewer #4: No

---

## [Author Response · Author response to Decision Letter 0]

7 Nov 2022

Point-by-Point Responses to the Editor and Reviewers

Journal Requirements:

Q1:

Reply: We have ensured that the paper meets these PLOS ONE style requirements. 

Q2:

a) Did participants provide their written or verbal informed consent to participate in this study?

Reply: All patients consented in writing. This point has been added to the revised manuscript. (Line 89)

Q3:

Reply: Our research was not funded by any specific grant from any funding agency in the public, commercial, or not-for-profit sectors. The previous statement (“The funders had no role in study design, data collection and analysis, decision to publish, or preparation of the manuscript”) was made in error. We apologize for this mistake and have included the statement “The authors received no specific funding for this work” in our revised cover letter (Point 6). 

Q4:

Reply: We have added the captions for Supplementary Tables S1–S3 to the end of the manuscript (Lines 591–597). Changes to the references were not required.

Reviewer comments:

General note to the editor and reviewers: We deleted the Neiter et al. reference (Original Ref 21), which did not find a relationship between preoperative CCT and postoperative VA, because (i) half of that cohort was treated by DSAEK, and (ii) we found three more pertinent studies on the preoperative CCT:post-DMEK VA relationship during revision (Peraza-Nieves et al. 2017; Schrittenlocher et al. 2019; Machalinska et al. 2021). As a result, four papers support our finding that preoperative CCT does not correlate with postoperative VA while one (Peraza-Nieves et al.) supports the relationship observed by Brockmann et al. We created a table that summarizes this literature (New Table 1). Consequently, Original Tables 1–4 are now called Tables 2–5, respectively.

We also rewrote parts of the Abstract, Introduction, Discussion, and Conclusions to address reviewer questions about why we examined postoperative CCT, given that this variable is not clinically useful for predicting DMEK visual outcomes. 

New references:

Machalińska A, Kuligowska A, Kaleta K, Kuśmierz-Wojtasik M, Safranow K. Changes in Corneal Parameters after DMEK Surgery: A Swept-Source Imaging Analysis at 12-Month Follow-Up Time. J Ophthalmol. 2021 Jul 21;2021:3055722. doi: 10.1155/2021/3055722.

Schrittenlocher S, Bachmann B, Tiurbe AM, Tuac O, Velten K, Schmidt D, Cursiefen C. Impact of preoperative visual acuity on Descemet Membrane Endothelial Keratoplasty (DMEK) outcome. Graefes Arch Clin Exp Ophthalmol. 2019 Feb;257(2):321-329. doi: 10.1007/s00417-018-4193-4. 

Peraza-Nieves J, Baydoun L, Dapena I, Ilyas A, Frank LE, Luceri S, Ham L, Oellerich S, Melles GRJ. Two-Year Clinical Outcome of 500 Consecutive Cases Undergoing Descemet Membrane Endothelial Keratoplasty. Cornea. 2017 Jun;36(6):655-660. doi: 10.1097/ICO.0000000000001176

Reviewer #1: 

Q1:

Authors should have a plenty of data.

They should analyze and rewrite the paper.

Reply: Thank you very much for the time you spent reading our manuscript and for your helpful comments. We have addressed them to the best of our ability. 

Q2:

1) There is no description of included criteria despite the prospective fashion.(Line102-107)

Reply: Our study was retrospective but the data themselves were recorded into the medical database prospectively (i.e. immediately after each DMEK). This point was made in Original Line 94 (now New Line 94). 

The inclusion and exclusion criteria were:

Inclusion: (1) Consecutive adult (≥18 years) patients (2) with FECD who (3) underwent DMEK alone (if they were pseudophakic) or triple-DMEK (cataract surgery followed by DMEK) between October 2015 and April 2021 and (4) were followed for at least 12 months. 

Exclusion from patient selection: (1) Another ophthalmological disease that could interfere with visual acuity (e.g. retinal detachment, central retinal vein occlusion, severe age-related macular degeneration, or advanced glaucoma), (2) a history of previous corneal graft, (3) preoperative BSCVA data were missing, (4) the patient was lost to follow-up, and (5) long-standing bullous keratopathy cases (these were not included because we use DSAEK to treat such patients).

Exclusion from analyses: (1) Second operated eye in bilateral DMEK cases, and (2) eyes with graft failure during the first postoperative year. 

Except for the last exclusion criterion (graft failure), these points were included in the Methods section of the original manuscript (Original Lines 94¬–103). The numbers of eyes that met these criteria (including graft failure) were indicated in the flow diagram in Fig 1 and the beginning of the Results section (Original Lines 180–189).

To address this comment, we added the missing exclusion-from-analysis criterion (graft failure) to the Methods and made the related text clearer in general, as shown in Lines 94–106.

Q3:

2)It doesn't make sense if you evaluate the preoperative factor, and postoperative factor.(Line164-174)

Reply: We agree that our analysis of postoperative factors was confusing, this point was also raised by another reviewer. 

The reason we included these analyses is because they may explain why Brockmann et al. found a correlation between preoperative CCT and postoperative VA whereas we and others did not. 

Relevant background to answering this question is a paper that we recently published (Perone et al., 2021=Ref 30 in revised manuscript) on a vexed question in the literature, namely, why does preoperative (graft) central graft thickness (CGT) in DSAEK shape final VA in some studies but not others? In Perone et al. (2021), we observed that while preoperative CGT did not predict postoperative BSCVA, postoperative CGT did. We speculated that the interstudy variability in the ability of preoperative CGT to determine postoperative VA may reflect inaccuracies or variation in the preoperative measurements, which are shaped by many factors (e.g. graft culture time). These factors disappear after surgery, thus allowing postoperative CGT to predict VA.

Therefore, when we observed in the current study that preoperative CCT did not correlate with final BSCVA after DMEK, we examined whether postoperative CCT did have a relationship with final BSCVA. Indeed, it did (see New Table 5, which used to be Old Table 4). Therefore, we proposed that inaccuracies in preoperative CCT measurements due to various factors (e.g. the fulminant corneal disease) may blur the relationship between this variable and postoperative VA, whereas postoperative CCT measurements are not affected by such factors.

To address this issue, we have clarified this aspect of the paper in the Abstract (New Lines 44–47), Introduction (New Lines 79–81), Discussion (New Lines 269–311), and Conclusions (New Lines 356–357). 

Q4:

Please ask the specialist of statistics, and reconsider the methodology.

Please add the statistic specialist in the acknowledgement.

Reply: The biostatistician is Dr. Goetz, the head of the Clinical Research Support Unit of our hospital. He is an author on our paper.

We did not conduct a multivariate analysis for the purposes of this study because we wanted to see whether we could replicate the findings of Brockmann et al., who conducted univariate analyses to obtain their preoperative CCT threshold of 625 µm. 

Q5:

3)It doesn't make sense if you compare the CCT over 625, and under 625 um, since the mean CCT you showed was 695.

Please justify the use of 625. (Line 219-222)

Reply: The threshold of 625 µm that we used was derived from Brockmann et al.: the purpose of our study was to determine whether that preoperative CCT threshold could indeed identify patients whose VA recovers less well after DMEK than patients with thinner preoperative CCTs.

In our study, the median preoperative CCT was 626 µm (see New Table 2). We did not conduct analyses with this potential cut-off because it was so close to the threshold of Brockmann et al. (625 µm).

Q6:

4) The discussion explaining the correlation between postoperative CCT and the BSVCA at 12months.

You can evaluate HOAs more since you used Pentacam. Please evaluate more using HOAs.

Please refer to the following paper. (https://pubmed.ncbi.nlm.nih.gov/33052928/)

Reply: We read the paper by Hayashi et al. with interest and certainly agree that HOA can affect visual recovery and is a postoperative variable of considerable interest. However, we were interested in whether a preoperative variable, CCT, could predict final VA (as suggested by the recent paper of Brockmann et al.), since this would help us identify patients who should be encouraged to undergo DMEK sooner rather than later.

Q7:

Anyway, the current version is not proper in terms of statics, or logics.

Reply: We hope our revision and answers here have allayed your concerns. We appreciate the time you spent on our manuscript and feel that addressing your comments has significantly improved our manuscript.

Reviewer #2: 

Nicely written no major concerns should add to the literature that pre op cct should not be a variable to consider. Use bcva and patient function as criteria always issues with retrospective study but not much else published

Reply: Thank you very much for your time and positive comments!

Reviewer #3: 

Dear Authors,

thank you very much for this interesting paper about pre- and postoperative CCT in DMEK and the influence on BCVA.

Reply: Thank you very much for your time and your valuable comments. We have addressed them to the best of our ability and feel they have significantly improved our manuscript.

Q1:

In the abstract it is stated that extremely decompensated eyes were not included. This is not present in the methods section and should be added. However, I would rather recommend to include those eyes, since this might have an impact on the prediction with preoperative CCT.

Reply: It is our general policy to treat such eyes with DSAEK because it is more complicated to perform DMEK in such eyes, which increases the risk of graft detachment and other surgery-related complications. Indeed, many centers use penetrating keratoplasty for such eyes for this reason.

To address this comment, we added the term ‘extremely decompensated’ to the Methods text. (New Line 105)

Reviewer #4: 

Q1:

In the introduction, the authors describe the recent studies that correlate prep CCT and postoperative vision. However, the primary conclusions are written in a confusing manner. Predictive factors are of course relevant when they are identified preoperatively or perioperatively. The authors should focus their analysis on the preoperative factors.

Reply: We agree that our analysis of postoperative factors was confusing, this point was also raised by another reviewer. 

The reason we included these analyses is because they may explain why Brockmann et al. found a correlation between preoperative CCT and postoperative VA whereas we and others did not. 

Relevant background to answering this question is a paper that we recently published (Perone et al., 2021=Ref 30 in revised manuscript) on a vexed question in the literature, namely, why does preoperative (graft) central graft thickness (CGT) in DSAEK shape final VA in some studies but not others? In Perone et al. (2021), we observed that while preoperative CGT did not predict postoperative BSCVA, postoperative CGT did. We speculated that the interstudy variability in the ability of preoperative CGT to determine postoperative VA may reflect inaccuracies or variation in the preoperative measurements, which are shaped by many factors (e.g. graft culture time). These factors disappear after surgery, thus allowing postoperative CGT to predict VA.

Therefore, when we observed in the current study that preoperative CCT did not correlate with final BSCVA after DMEK, we examined whether postoperative CCT did have a relationship with final BSCVA. Indeed, it did (see New Table 5, which used to be Old Table 4). Therefore, we proposed that inaccuracies in preoperative CCT measurements due to various factors (e.g. the fulminant corneal disease) may blur the relationship between this variable and postoperative VA, whereas postoperative CCT measurements are not affected by such factors.

To address this issue, we have clarified this aspect of the paper in the Abstract (New Lines 44–47), Introduction (New Lines 79–81), Discussion (New Lines 269–311), and Conclusions (New Lines 356–357). 

Q2:

It would also be relevant to determine if postoperative CCT correlates to preop parameters.

Reply: We conducted these analyses during the revision but did not find that any of the preoperative variables correlated with postoperative CCT (see Table below). We therefore decided to keep the focus of our paper on the relationship between CCT and BSCVA, and did not mention these data in the revised manuscript.

Variable 12 month CCT p*

 Median (IQR) r (95% CI) 

Patient age -0.03 (-0.21; 0.15) 0.74

Patient sex 0.47

 Female 524 (508–560) 

 Male 531 (515–558) 

Preoperative VA -0.10 (-0.28; 0.07) 0.25

Preoperative graft ECD -0.11 (-0.28 ; 0.07) 0.24

Donor age 0.05 (-0.13 ; 0.23) 0.58

Triple DMEK 0.38

 Yes 532 (512 ; 568) 

 No 527 (508 ; 555) 

*Wilcoxon tests or Spearman correlations

Q3:

In light of findings that preop CCT does not affect postoperative vision. Compare with other studies correlating CCT and postoperative vision such as:

1. https://pubmed.ncbi.nlm.nih.gov/30339062/ Brockman et al. 2019

2. https://pubmed.ncbi.nlm.nih.gov/33591041/ Gundlach et al. 2021

3. https://pubmed.ncbi.nlm.nih.gov/34336256/ Machalinksa et al. 2021

4. https://www.ncbi.nlm.nih.gov/pmc/articles/PMC8786756/ Ademmer et al. 2022

5. https://pubmed.ncbi.nlm.nih.gov/27986425/ Schaub et al. 2017

Revise accordingly.

Reply: Thank you very much for these five references, which we read carefully. 

The first paper https://pubmed.ncbi.nlm.nih.gov/30339062/ is by Brockmann et al., whose work we wished to reproduce in our study because their preoperative CCT cut-off of 625 µm would be useful for judging when to conduct DMEK. Unfortunately, we did not find that this cut-off was useful in our cohort; in fact, we found that preoperative CCT did not correlate with final visual acuity.

The fifth paper https://pubmed.ncbi.nlm.nih.gov/27986425/ by Schaub et al. was cited in the Introduction (Original Lines 75–77) and Discussion (Original Lines 262–264) of our paper (Original Ref 22). Like us, they did not find a correlation between preoperative CCT and postoperative BSCVA.

The third paper https://pubmed.ncbi.nlm.nih.gov/34336256/ by Machalinska et al. was also particularly relevant. This study relates to 24 eyes with FECD or PBK that underwent DMEK and serial ophthalmological measurements at baseline and postoperative months 1, 3, 6, and 12. They did not find significant correlations between preoperative CCT and change in BSCVA (relative to baseline) at any of the postoperative time points (r=0.20–0.27, all p>0.05). This is consistent with our findings. 

The second paper https://pubmed.ncbi.nlm.nih.gov/30339062/ by Gundlach et al. is on 50 eyes with FECD and PBK that underwent DMEK. They found that preoperative contrast sensitivity correlated with preoperative CCT. However, correlations between CCT and BSCVA were not assessed.

The fourth reference https://www.ncbi.nlm.nih.gov/pmc/articles/PMC8786756/ is by Ademmer et al., who showed that preoperative BSCVA and CCT did not correlate with subjective visual complaints, as measured with a questionnaire. However, they did not mention correlations between preoperative CCT and postoperative BSCVA.

Given that we missed the Machalinska et al. reference, we conducted another extensive Google literature search. We found two other studies that examined the relationship between preoperative CCT and postoperative VA, as follows:

Schrittenlocher et al (2019) assessed 1084 eyes that underwent DMEK for FECD and found no correlation between preoperative CCT and 12-month BSCVA.

Peraza-Nieves et al. (2017) examined 393 eyes that underwent DMEK for FECD or PBK and found that preoperative CCT predicted 24-month visual acuity (p<0.0001) on multiple regression analysis. 

Therefore, to address this comment, we created a literature-summarizing table (New Table 1) and added these references and texts regarding the papers to the Introduction (New Lines 75–77) and Discussion (New Lines 269–282, 304–307).

---

## [Decision Letter · Decision Letter 1]

21 Feb 2023

Postoperative, but not preoperative, central corneal thickness correlates with the postoperative visual outcomes of Descemet membrane endothelial keratoplasty

PONE-D-22-27294R1

Dear Dr. Perone,

We’re pleased to inform you that your manuscript has been judged scientifically suitable for publication and will be formally accepted for publication once it meets all outstanding technical requirements.

You may disregard reviewer #4s comments.

Kind regards,

Michael Mimouni

Academic Editor

PLOS ONE

Additional Editor Comments (optional):

Reviewers' comments:

Reviewer's Responses to Questions

**Comments to the Author**

1. If the authors have adequately addressed your comments raised in a previous round of review and you feel that this manuscript is now acceptable for publication, you may indicate that here to bypass the “Comments to the Author” section, enter your conflict of interest statement in the “Confidential to Editor” section, and submit your "Accept" recommendation.

Reviewer #1: All comments have been addressed

Reviewer #3: All comments have been addressed

Reviewer #4: (No Response)

2. Is the manuscript technically sound, and do the data support the conclusions?

Reviewer #1: Yes

Reviewer #3: Yes

Reviewer #4: No

3. Has the statistical analysis been performed appropriately and rigorously? 

Reviewer #1: Yes

Reviewer #3: Yes

Reviewer #4: No

4. Have the authors made all data underlying the findings in their manuscript fully available?

Reviewer #1: Yes

Reviewer #3: Yes

Reviewer #4: No

5. Is the manuscript presented in an intelligible fashion and written in standard English?

Reviewer #1: (No Response)

Reviewer #3: Yes

Reviewer #4: No

6. Review Comments to the Author

Reviewer #1: The manuscript has been well revised.

There is no additional comment. However, further study may be necessary to know the truth in terms of the correlation between preoperative factors and postoperative outcomes.

Reviewer #3: Dear Authors,

thank you for adressing the many comments by all reviewers. I have no further requests.

Kind regards!

Reviewer #4: authors failed to address previous comments. the primary conclusions are written in a confusing manner. Predictive factors are of course relevant when they are identified preoperatively or perioperatively. The authors should focus their analysis on the preoperative factors.

7. PLOS authors have the option to publish the peer review history of their article (what does this mean?). If published, this will include your full peer review and any attached files.

Reviewer #1: No

Reviewer #3: No

Reviewer #4: No

---

## [Editor Report · Acceptance letter]

24 Feb 2023

PONE-D-22-27294R1 

Postoperative, but not preoperative, central corneal thickness correlates with the postoperative visual outcomes of Descemet membrane endothelial keratoplasty 

Dear Dr. Perone:

I'm pleased to inform you that your manuscript has been deemed suitable for publication in PLOS ONE. Congratulations! Your manuscript is now with our production department. 

Kind regards, 

on behalf of

Dr. Michael Mimouni 

Academic Editor

PLOS ONE